# Group Cognition Learning: Making Everything Better Through Governed Two-Stage Agents Collaboration

**Chunlei Meng** [1]   **Pengbin Feng** [2]   **Rong Fu** [3]   **Hoi Leong Lee** [4]   **Xiaojing Du** [5]   **Zhaolu Kang** [6]   **Zeyu Zhang** [7]
**Weilin Zhou** [8]   **Chun Ouyang*** [1]   **Zhongxue Gan** [1]

## Abstract

Centralized multimodal learning commonly compresses language, acoustic, and visual signals into a single fused representation for prediction. While effective, this paradigm suffers from two limitations: modality dominance, where optimization gravitates towards the path of least resistance, ignoring weaker but informative modalities, and spurious modality coupling, where models overfit to incidental cross-modal correlations. To address these, we propose **Group Cognition Learning (GCL)**, a governed collaboration paradigm that applies a two-stage protocol after modality-specific encoding. In Stage 1 (Selective Interaction), a Routing Agent proposes directed interaction routes, and an Auditing Agent assigns sample-wise gates to emphasize exchanges that yield positive marginal predictive gain while suppressing redundant coupling. In Stage 2 (Consensus Formation), a Public-Factor Agent maintains an explicit shared factor, and an Aggregation Agent produces the final prediction through contribution-aware weighting while keeping each modality representation as a specialization channel. Extensive experiments on CMU-MOSI, CMU-MOSEI, and MIntRec demonstrate that GCL mitigates dominance and coupling, establishing state-of-the-art results across both regression and classification benchmarks. Analysis experiments further demonstrate the effectiveness of the design.

## 1 Introduction

Multimodal learning integrates heterogeneous evidence from language, acoustic prosody, and visual expressions. This problem is central to multimodal affective computing, where models infer sentiment polarity and intensity from videos (Meng et al., 2026d;c), and it also arises in multimodal intent understanding, where textual and audio-visual cues jointly indicate user intent (Wu et al., 2025; Meng et al., 2025; 2026b). A high-quality multimodal learning framework must reconcile a structural tension: it should preserve modality specialization because each modality contains distinct label-relevant cues, while it should acquire complementary evidence because the label is often supported by multiple weak but consistent signals (Hazarika et al., 2020; Meng et al., 2026a).

Existing methods typically address this integration through centralized fusion. Early fusion operators such as Tensor Fusion Network and Low-rank Multimodal Fusion explicitly model cross-modal interactions (Zadeh et al., 2017; Liu et al., 2018), while attention-based and transformer-style designs strengthen sequence-level fusion and alignment, including multimodal transformers for unaligned sequences and pretrained transformer fusion (Tsai et al., 2019; Rahman et al., 2020; Yu et al., 2021). Recent multimodal affective computing methods further push end-to-end fusion in the wild (Wu et al., 2025; Fang et al., 2025; Wang et al., 2025). Beyond fusion operators, representation-structuring methods aim to preserve modality-specific cues by separating shared and private factors or disentangling multimodal information, including modality-invariant and modality-specific decomposition (Hazarika et al., 2020), contrastive feature decomposition (Yang et al., 2023), and multimodal information disentanglement (Dai et al., 2024). Related lines also pursue disentanglement for multimodal emotion and sentiment modeling (Meng et al., 2026d; Li et al., 2023).

Despite these advances, most methods implicitly assume that a unified optimization loop will naturally discover optimal interaction patterns. In practice, interaction is optimized only through the final task loss, without explicit signals that distinguish beneficial exchange from redundant co-activation (Wei et al., 2025b). Gradient-based training

---

[1]The College of Intelligent Robotics and Advanced Manufacturing, Fudan University [2]University of Southern California [3]University of Macau [4]Universiti Malaysia Perlis [5]Adelaide University [6]Peking University [7]The Australian National University [8]Xinjiang University. Correspondence to: Chun Ouyang <c_oy@fudan.edu.cn; clmeng23@m.fudan.edu.cn>.

*Proceedings of the 43rd International Conference on Machine Learning*, Seoul, South Korea. PMLR 306, 2026. Copyright 2026 by the author(s).

therefore tends to lock onto the easiest predictive pathway early and to couple interaction learning with representation learning, so the resulting interaction patterns become highly sensitive to early dynamics and modality reliability (Wei et al., 2025b; Guo et al., 2024). This sensitivity turns implicit interaction learning into a systematic failure mode, where shortcuts are reinforced even when they are incidental for the label, and specialization is gradually eroded (Yang et al., 2024; Wei et al., 2025b). Prior works provide consistent evidence for this diagnosis. Optimization interventions rebalance modality contributions to mitigate biased reliance (Guo et al., 2024; Yang et al., 2024), expert or routing style designs introduce adaptive collaboration patterns instead of relying on centralized fusion to self-correct (Fang et al., 2025; Chen et al., 2024), and disentanglement based methods constrain shared versus modality-specific factors to counter co-adaptation and redundant correlations (Hazarika et al., 2020; Yang et al., 2023; Dai et al., 2024; Yang et al., 2022; Sun et al., 2024a; Li et al., 2023). Yet these approaches still lack an explicit governance mechanism that audits which interactions should be admitted and which dependencies should be suppressed at the interaction process level. Under such weakly governed interaction learning, two limitations recur. First, modality dominance is an optimization consequence. With a single fused pathway, gradients concentrate on the modality that reduces loss fastest, yielding a low resistance shortcut that under trains other modalities and makes prediction brittle under noise. Second, spurious modality coupling is a co-adaptation effect. End-to-end fusion rewards feature alignment even when correlations are incidental, pushing modalities to encode overlapping content and increasing sensitivity to shifts that break these co occurrences (Guo et al., 2024; Yang et al., 2024). If not addressed, models may look strong in distribution but degrade under mild shifts, lose per modality interpretability, and behave unstably with corrupted evidence.

We address these limitations at their source by defining multimodal collaboration as protocol governed interaction learning. Formally, we propose a novel **Group Cognition Learning (GCL)** paradigm which learns an interaction protocol that maps modality encodings to a set of admissible directed exchanges and a consensus operator, so the system governs the interaction process rather than only learning fusion weights. Concretely, GCL implements a two stage protocol with four agents that have explicit responsibilities. **In Stage 1, Selective Interaction**, a Routing Agent proposes directed routes among modalities and an Auditing Agent performs sample-wise admission by predicting marginal predictive gain and penalizing redundancy, so exchanges are encouraged only when they improve prediction and are discouraged when they increase coupling. **In Stage 2, Consensus Formation**, a Public-Factor Agent maintains an explicit shared factor that captures cross modality com-

mon signals, and an Aggregation Agent forms the final decision using contribution-aware weighting conditioned on the shared factor, while keeping each modality representation as a specialization channel.

- We propose GCL, a two-stage governed collaboration paradigm that targets modality dominance and spurious modality coupling by explicitly regulating cross-modal interaction rather than leaving it implicit in centralized fusion.

- In Stage 1, we design Routing Agent and Auditing Agent to realize a marginal-gain-driven selective exchange and redundancy-governed interaction.

- In Stage 2, we design Public-Factor Agent and Aggregation Agent to separate shared signals from private specialization and to form consensus without collapsing modalities into a single entangled feature.

## 2 Related Work

**Multimodal Learning.** Early designs explicitly model interactions using tensor or low-rank operators (Zadeh et al., 2017; Liu et al., 2018), while transformer-based architectures strengthen sequence-level alignment for unaligned signals (Tsai et al., 2019; Rahman et al., 2020). Beyond fusion, representation structuring methods aim to preserve modality-specific cues via disentanglement (Hazarika et al., 2020; Yang et al., 2022). However, these methods typically optimize interaction implicitly through a unified task loss. Consequently, they often fail to prevent modality dominance and spurious coupling, eroding specialization and robustness under noise (Guo et al., 2024; Yang et al., 2024; Wei et al., 2025b).

**Interaction Failures and Interventions.** Recent work recognizes that end-to-end training often discovers unreliable patterns, locking onto easy predictive pathways or spurious shortcuts due to implicit gradient dynamics (Wei et al., 2025b; Guo et al., 2024; Yang et al., 2024). To mitigate this, one line introduces optimization interventions to rebalance modality contributions (Guo et al., 2024; Yang et al., 2024), while routing-style designs learn adaptive collaboration patterns (Fang et al., 2025). Despite these advances, existing methods lack a transparent governance mechanism to explicitly audit interaction validity and suppress redundant dependencies at the process level.

## 3 Method

### 3.1 Problem Formulation

Let $\mathcal{M} = \{l, a, v\}$ denote the set of modalities (language, acoustic, visual). For a sample $x = \{x^m\}_{m \in \mathcal{M}}$, we aim to predict a target $y$, where $y \in \mathbb{R}$ for regression or $y \in$

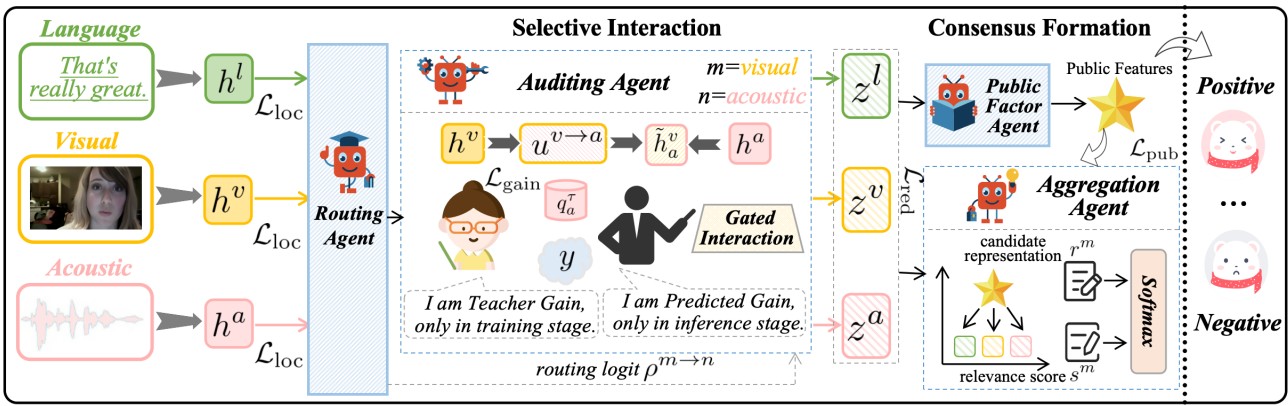

*Figure 1.* Overview of the GCL architecture. The paradigm implements a two-stage governed collaboration protocol. Governed Interaction (Stage 1) uses Routing and Auditing agents to regulate cross-modal exchange based on marginal predictive gain. Consensus Formation (Stage 2) employs Public-Factor and Aggregation agents to synthesize predictions anchored by a shared semantic factor.

$\{1, \ldots, C\}$ for classification. Each modality is processed by a specific encoder to yield initial representations. The core objective of GCL is to transform these independent representations into a joint prediction through a governed process that maximizes task information while minimizing redundant cross-modal information.

### 3.2 Protocol-Governed Interaction Learning Framework

We reformulate multi-modal learning not as a static fusion operation, but as a dynamic, protocol-governed interaction process. Unlike conventional approaches that implicitly assume a fixed connectivity graph across modalities, GCL imposes an explicit governance layer atop the modality-specific encoders. This framework operates under the premise that effective collaboration requires regulating the information flow to ensure it is both purpose-driven and audit-compliant.

As illustrated in Figure 1, the architecture is systematized into a two-stage protocol, orchestrating four specialized agents to balance the trade-off between modality interaction and independence. The first stage, designated as Selective Interaction, governs the topology of information exchange. By evaluating the marginal predictive gain, it transitions the system from a fully connected graph to a sample-specific sparse structure, admitting only those interactions that demonstrably contribute to the learning objective. The second stage, Consensus Formation, synthesizes the final decision. It disentangles cross-modal commonalities into an explicit public factor while maintaining private specialization channels, thereby preventing feature collapse during integration. Through this hierarchical design, GCL transforms the opaque fusion weights of traditional models into transparent control signals. Each agent within this protocol is assigned a distinct functional role and is constrained

by specific supervision signals, ensuring that the collective behavior of the system remains interpretable and aligned with the overarching governance objectives.

### 3.3 Stage 1: Selective Interaction

Conventional multi-modal learning often relies on holistic fusion strategies, such as concatenation or dense cross-attention, which implicitly assume that all cross-modal connections are beneficial. However, this indiscriminate interaction frequently leads to the propagation of noise and the accumulation of redundant information, particularly when modalities are misaligned or uninformative. To address this, Stage 1 introduces Selective Interaction, a governance protocol designed to transform cross-modal exchange from a passive, fixed-topology process into an active, sample-adaptive transaction. The core objective is to regulate information flow based on utility: interaction pathways are established if and only if they provide a measurable predictive gain. This architecture is materialized through two cooperative agents: the **Routing Agent**, which proposes candidate interaction pathways, and the **Auditing Agent**, which rigorously evaluates and sanctions these proposals. Together, they enforce a sample-specific topology that maximizes information gain while minimizing redundancy.

**Routing Agent: Pathway Proposal.** The Routing Agent acts as the initiator of the interaction protocol. Its primary responsibility is to identify potential directions of information transfer and package the semantic content for exchange. Structurally, it comprises a route proposer and a message mapper. For every directed pair $(m \rightarrow n)$ with $m \neq n$, the agent computes a routing logit $\rho^{m \rightarrow n}$ to signal the intent to interact, and constructs a compact message vector $u^{m \rightarrow n}$ to constrain the channel capacity:

$$\rho^{m \rightarrow n} = g_r^{m \rightarrow n} r(h^l, h^a, h^v), \quad u^{m \rightarrow n} = \psi^{m \rightarrow n}(h^m). \quad (1)$$

In implementation, $g_r^{m \rightarrow n}$ utilizes a lightweight MLP over

the concatenated global context to assess topological feasibility, while $\psi^{m \to n}$ projects the source modality into a bottlenecked latent space. This intentional separation decouples the decision to interact from the content of the interaction, ensuring flexible routing policies without creating high-dimensional entanglements.

**Auditing Agent: Evaluation and Gated Execution.** The Auditing Agent functions as the gatekeeper, tasked with assessing the admissibility of the proposed routes. Unlike static gating mechanisms, the Auditing Agent relies on a rigorous utility metric: the marginal predictive gain.

*Gain Estimation and Admission.* To determine whether a message $u^{m \to n}$ should be integrated into the recipient $n$, the agent estimates the potential reduction in task loss. During training, we compute a "teacher gain" $\Delta^{m \to n}$ by comparing the loss of a local task head $q_n^\tau$ before and after tentatively integrating the message, defined as:

$$\Delta^{m \to n} = \ell_\tau\big(q_n^\tau(h^n), y\big) - \ell_\tau\big(q_n^\tau(\tilde{h}_m^n), y\big). \quad (2)$$

where $\tilde{h}_m^n = h^n + \phi^{m \to n}(h^n, u^{m \to n})$ denotes the tentatively updated representation and $\phi^{m \to n}$ fuses the message with the target modality representation. During inference, since label supervision is unavailable, the agent learns a parameterized gain predictor to approximate the teacher gain, defined as

$$\hat{\Delta}^{m \to n} = g_g^{m \to n}(h^n, u^{m \to n}). \quad (3)$$

The final admission gate $\alpha^{m \to n}$ is then derived by combining the routing intent with the gain ($\tilde{\Delta}$ is $\Delta^{m \to n}$ during training and $\hat{\Delta}^{m \to n}$ during inference):

$$\alpha^{m \to n} = \frac{\exp(\rho^{m \to n})}{\sum_{j \in \mathcal{M} \setminus \{n\}} \exp(\rho^{j \to n})} \cdot \sigma_\kappa\big(\tilde{\Delta}^{m \to n}\big). \quad (4)$$

Here, the routing logits is normalized across potential senders, and the sigmoid function $\sigma$ (scaled by temperature $\kappa$) converts the predicted gain into a probabilistic gate. This ensures that the protocol relies on the ground-truth teacher gain for supervision during training, while seamlessly transitioning to the predicted gain $\hat{\Delta}^{m \to n}$ for inference. *Gated Integration.* Upon admission, the Auditing Agent executes the information exchange via a gated residual update:

$$z^n = h^n + \sum_{m \in \mathcal{M} \setminus \{n\}} \alpha^{m \to n} \cdot \phi^{m \to n}(h^n, u^{m \to n}). \quad (5)$$

The resulting representations $\{z^n\}$ serve as refined, specialized channels for the subsequent stage.

**Regularization: Redundancy Control and Gain Alignment.** To prevent the selective interaction from collapsing into trivial co-adaptation, we impose two critical constraints. First, we penalize redundant coupling among the updated modality channels using a contrastive redundancy score $D_{\text{red}}(\cdot, \cdot)$. Different from the conventional InfoNCE objective that minimizes a cross-entropy loss to increase positive-pair alignment, our score measures how easily the matched cross-modal channels can be identified from each other within a batch.

$$\mathcal{L}_{\text{red}} = \sum_{m < n} D(z^m, z^n). \quad (6)$$

Minimizing $\mathcal{L}_{\text{red}}$ enforces orthogonality between the specialized channels, discouraging the model from learning shortcut solutions based on incidental correlations. Second, to align the gate learning with actual utility, we introduce a gain alignment loss:

$$\mathcal{L}_{\text{gain}} = - \sum_{n \in \mathcal{M}} \sum_{m \in \mathcal{M} \setminus \{n\}} \alpha^{m \to n} \, \text{stopgrad}(\Delta^{m \to n}). \quad (7)$$

This objective encourages the agent to open gates (large $\alpha^{m \to n}$) when the teacher gain $\Delta^{m \to n}$ is positive, and to close gates (small $\alpha^{m \to n}$) when the gain is negative. The stopgrad$(\cdot)$ operator prevents gradients from flowing through the teacher gain, ensuring that $\Delta^{m \to n}$ serves purely as a supervision signal, thereby stabilizing the governance mechanism and ensuring that interactions are driven by genuine informativeness.

### 3.4 Stage 2: Consensus Formation

Following the selective interaction, the objective of Stage 2 is to synthesize a unified prediction by reconciling heterogeneous modality-specific evidence with cross-modal semantic consensus. Unlike naive fusion strategies that often suffer from feature collapse or modality dominance, this stage enforces a structured integration protocol designed to preserve the distinctiveness of private representations while leveraging shared contexts. Specifically, we decompose the consensus formation process into two cooperative roles: extracting a global semantic invariant and dynamically calibrating individual modality contributions based on this invariant. This architecture is materialized by two agents: the **Public-Factor Agent**, which distills the cross-modal commonalities, and the **Aggregation Agent**, which executes the context-aware fusion.

**Public-Factor Agent.** To distill the semantic consensus inherent across modalities, we introduce the Public Factor Agent, parameterized by a shared extractor $g_p$. The module aggregates the refined specialization channels $\{z^l, z^a, z^v\}$ to synthesize an explicit public factor $c$. The projection is defined as: $c = g_p(z^l, z^a, z^v)$. where $g_p$ is instantiated as a permutation-invariant operator, such as a symmetric attention mechanism or global pooling followed by an MLP. This design ensures the extracted commonality is robust to input ordering and effectively decouples shared semantics

from modality-exclusive evidence, thereby preventing the entanglement of private representations. To further guarantee the semantic validity of the public factor, we impose an auxiliary supervision signal on $c$ within the overall objective, enforcing it to be predictive of the target while remaining structurally distinct from the private subspaces.

**Aggregation Agent.** To synthesize a unified representation from the disentangled factors, the Aggregation Agent orchestrates an adaptive fusion mechanism. Formally, for each modality $m$, we employ a proposal network $\eta_m$ and a gating network $g_a^m$ to generate a candidate representation $r^m = \eta_m(z^m, c)$ and an unnormalized relevance score $s^m = g_a^m(z^m, c)$. These scores are subsequently normalized via $\pi^m = \mathrm{softmax}(\{s^m\}_{m \in \mathcal{M}})$ to dynamically determine the contribution of each modality based on sample-specific evidence. Consequently, the final task prediction is derived by aggregating these weighted proposals into a consensus vector $r_{con}$, formulated as:

$$\hat{o} = g^\tau(r_{con}, c), \quad \text{where} \quad r_{con} = \sum_{m \in \mathcal{M}} \pi^m r^m. \quad (8)$$

Functionally, conditioning the gating mechanism on $c$ incorporates the global semantic context into the weight allocation process. By deriving the consensus $r_{con}$ through a weighted summation of distinct proposals $r^m$, the architecture maintains the independence of private feature channels prior to the final aggregation step.

### 3.5 Overall Objective

We train the governance protocol and modality-specific representations jointly under a unified supervised objective. Let $\hat{o}$ denote the final consensus prediction in Eq. (8). The primary task loss is

$$\mathcal{L}_{\mathrm{task}} = \mathbb{E}_{(x,y) \sim \mathcal{D}}\big[\ell_\tau(\hat{o}, y)\big], \quad (9)$$

where $\ell_\tau$ is the task loss (e.g., cross-entropy for classification or squared error for regression).

To make the teacher gain $\Delta^{m \to n}$ in Stage 1 a reliable auditing signal, the local heads $\{q_m^\tau\}$ must provide meaningful estimates of modality-wise predictive risk before interaction. We therefore impose local supervision on unimodal representations:

$$\mathcal{L}_{\mathrm{loc}} = \sum_{m \in \mathcal{M}} \mathbb{E}_{(x,y) \sim \mathcal{D}}\big[\ell_\tau(q_m^\tau(h^m), y)\big]. \quad (10)$$

This term stabilizes the gain supervision and prevents private channels from collapsing into non-predictive features.

We also stabilize the public factor $c$ as an explicit shared anchor by an auxiliary head $\hat{o}_c = g_c^\tau(c)$:

$$\mathcal{L}_{\mathrm{pub}} = \mathbb{E}_{(x,y) \sim \mathcal{D}}\big[\ell_\tau(\hat{o}_c, y)\big]. \quad (11)$$

The total objective is

$$\mathcal{L}_{\mathrm{total}} = \mathcal{L}_{\mathrm{task}} + \lambda_{\mathrm{loc}}\mathcal{L}_{\mathrm{loc}} + \lambda_{\mathrm{pub}}\mathcal{L}_{\mathrm{pub}} + \lambda_{\mathrm{gain}}\mathcal{L}_{\mathrm{gain}} + \lambda_{\mathrm{red}}\mathcal{L}_{\mathrm{red}}, \quad (12)$$

where $\lambda$ terms balance prediction accuracy and protocol governance.

## 4 Experiments

### 4.1 Experimental Settings

**Benchmarks.** We evaluate on three benchmarks: CMU-MOSI (Zadeh et al., 2016), CMU-MOSEI (Bagher Zadeh et al., 2018), and MIntRec (Zhang et al., 2022). For multimodal sentiment analysis, CMU-MOSI provides 2,199 video segments split into 1,284/229/686 for train/validation/test, and CMU-MOSEI provides 22,856 segments split into 16,326/1,871/4,659. Both datasets use a label space of $[-3, 3]$ for sentiment intensity. For multimodal intent recognition, MIntRec contains 2,224 instances across 20 intent classes following standard partitions.

**Metrics.** For sentiment analysis, we report Acc-2, Acc-7, F1, MAE, and Pearson Correlation (Corr). Acc-2 is computed by thresholding sentiment into two classes, while Acc-7 follows the standard seven-level discretization (Fang et al., 2025). For intent recognition, we report Accuracy, Precision, Recall, and F1.

**Implementation Details.** We implement GCL in PyTorch and train on an NVIDIA A100 GPU (32GB). We use Adam with batch size 128 and weight decay $1 \times 10^{-4}$. We employ early stopping with patience 6. We follow the official train/validation/test split and repeat each experiment five times with different random seeds. For each run, the best checkpoint is selected on the validation set, and the averaged test performance over five runs is reported.

### 4.2 Comparison with State-of-the-art Methods

Table 1 and Table 2 present the comparative evaluation against a comprehensive suite of baselines, ranging from early fusion-centric models to recent optimization-based approaches. On the multimodal sentiment analysis benchmarks, GCL establishes a new state-of-the-art across all reported metrics. Specifically on the CMU-MOSI dataset, our method achieves an MAE of 0.685 and a binary accuracy of 86.79%, surpassing the competitive EMOE baseline which records 0.710 MAE and 85.4% accuracy. This performance advantage is further corroborated on the large-scale CMU-MOSEI dataset, where GCL lowers the MAE to 0.520 compared to 0.536 for EMOE. Furthermore, the evaluation on the MIntRec dataset confirms the generalization capability of our framework beyond sentiment analysis. In the intent recognition task, GCL consistently outperforms strong predecessors including EMOE and TSDA, demon-

*Table 1.* Performance Comparison on MOSI and MOSEI. ↑ and ↓ indicate that higher or lower value is better. Bold means best.

| Models | CMU-MOSI | | | | | CMU-MOSEI | | | | |
|---|---|---|---|---|---|---|---|---|---|---|
| | MAE (↓) | Corr (↑) | Acc-2(%) | Acc-7(%) | F1(%) | MAE (↓) | Corr (↑) | Acc-2(%) | Acc-7(%) | F1(%) |
| TFN (Zadeh et al., 2017) | 0.947 | 0.673 | 77.99 | 31.9 | 77.95 | 0.572 | 0.714 | 78.5 | 51.6 | 78.96 |
| LMF (Liu et al., 2018) | 0.950 | 0.651 | 77.90 | 33.82 | 77.80 | 0.575 | 0.714 | 80.54 | 51.59 | 80.94 |
| MulT (Tsai et al., 2019) | 0.846 | 0.725 | 81.70 | 40.05 | 81.66 | 0.673 | 0.677 | 80.85 | 48.37 | 80.86 |
| PMR (Lv et al., 2021) | 0.895 | 0.689 | 79.88 | 40.60 | 79.83 | 0.645 | 0.689 | 81.57 | 48.88 | 81.56 |
| MISA (Hazarika et al., 2020) | 0.788 | 0.744 | 82.07 | 41.27 | 82.43 | 0.594 | 0.724 | 82.03 | 51.43 | 82.13 |
| Self-MM (Yu et al., 2021) | 0.765 | 0.764 | 82.88 | 42.03 | 83.04 | 0.576 | 0.732 | 82.43 | 52.68 | 82.47 |
| FDMER (Yang et al., 2022) | 0.760 | 0.777 | 83.01 | 42.88 | 83.22 | 0.571 | 0.743 | 83.88 | 53.21 | 83.35 |
| DMD (Li et al., 2023) | 0.744 | 0.788 | 83.24 | 43.88 | 83.55 | 0.561 | 0.758 | 84.17 | 54.18 | 83.88 |
| MCIS (Yang et al., 2024) | 0.756 | 0.783 | 84.02 | 43.58 | 83.85 | 0.557 | 0.747 | 84.97 | 53.85 | 84.34 |
| CGGM (Guo et al., 2024) | 0.747 | 0.798 | 84.43 | 43.21 | 84.13 | 0.551 | 0.761 | 83.90 | 53.47 | 84.14 |
| DEVA (Wu et al., 2025) | 0.730 | 0.787 | 84.40 | 46.33 | 84.45 | 0.541 | 0.769 | 83.17 | 52.28 | 83.75 |
| DLF (Wang et al., 2025) | 0.731 | 0.781 | 85.06 | 47.08 | 85.04 | 0.536 | 0.764 | 85.42 | 53.9 | 85.27 |
| EMOE (Fang et al., 2025) | 0.710 | - | 85.4 | 47.7 | 85.4 | 0.536 | - | 85.3 | 54.1 | 85.3 |
| TSDA (Meng et al., 2026d) | 0.695 | 0.795 | 86.3 | 48.6 | 86.2 | 0.529 | 0.773 | 86.3 | 54.9 | 85.9 |
| **GCL (Ours)** | **0.685** | **0.812** | **86.79** | **49.06** | **86.40** | **0.520** | **0.785** | **86.78** | **55.36** | **86.55** |

strating that the proposed protocol is robust across varying data distributions and task definitions.

We attribute these empirical gains to the explicit governance introduced by the interaction protocol. Unlike conventional fusion methods that risk overfitting to dominant modalities or accumulating noise through indiscriminate connectivity, the Selective Interaction mechanism in Stage 1 effectively filters out redundant cross-modal couplings before they propagate. This ensures that the information exchange is driven strictly by predictive utility rather than incidental correlation. Moreover, the Consensus Formation in Stage 2 enforces a structural separation between shared semantic signals and modality-specific evidence. This disentanglement allows the model to leverage cross-modal commonalities without compromising the unique discriminative power of individual private channels, resulting in representations that are both semantically rich and resistant to feature collapse.

*Table 2.* Performance comparison on MIntRec, Higher is better.

| Models | Acc | F1 | Pre. | Rec. |
|---|---|---|---|---|
| MAG-BERT (Rahman et al., 2020) | 70.34 | 68.19 | 68.31 | 69.36 |
| MuLT (Tsai et al., 2019) | 72.58 | 69.36 | 70.73 | 69.47 |
| MISA (Hazarika et al., 2020) | 72.36 | 70.57 | 71.24 | 70.41 |
| GsiT (Jin et al., 2025) | 72.60 | 69.40 | 69.40 | 70.10 |
| CAGC (Sun et al., 2024b) | 72.53 | 70.62 | 70.86 | 70.55 |
| EMOE (Fang et al., 2025) | 72.58 | 70.73 | 72.08 | 70.86 |
| TSDA (Meng et al., 2026d) | 72.59 | 70.68 | 71.97 | 70.75 |
| **GCL (Ours)** | **72.74** | **70.95** | **72.31** | **71.24** |

### 4.3 Ablation Studies

To rigorously validate the efficiency of the governance protocols and architectural components, we conduct comprehensive ablation studies on the CMU-MOSI, CMU-MOSEI, and MIntRec benchmarks. The experimental results (Ta-

*Table 3.* Ablation studies on the benchmarks.

| Methods | MOSI | | MOSEI | | MIntRec | |
|---|---|---|---|---|---|---|
| | MAE↓ | Acc$_7$ | MAE↓ | Acc$_7$ | Acc | F1 |
| **GCL (full)** | **0.685** | **49.06** | **0.520** | **55.36** | **72.74** | **70.95** |
| *Modality configurations* | | | | | | |
| *w/o* Language | 0.905 | 38.60 | 0.742 | 45.10 | 66.25 | 64.10 |
| *w/o* Acoustic | 0.699 | 48.25 | 0.532 | 54.25 | 71.95 | 70.20 |
| *w/o* Visual | 0.701 | 48.05 | 0.534 | 54.10 | 71.80 | 70.05 |
| only Language | 0.714 | 47.10 | 0.545 | 53.55 | 70.85 | 69.10 |
| only Acoustic | 0.980 | 35.40 | 0.810 | 41.80 | 61.30 | 58.10 |
| only Visual | 0.935 | 37.20 | 0.772 | 43.50 | 62.70 | 59.30 |
| *Stage 1: Selective Interaction* | | | | | | |
| *w/o* R.-Agent | 0.694 | 48.55 | 0.526 | 54.75 | 72.10 | 70.35 |
| *w/o* A.-Agent | 0.699 | 48.00 | 0.532 | 54.20 | 71.70 | 69.90 |
| Full exchange | 0.721 | 46.10 | 0.558 | 52.40 | 70.90 | 68.95 |
| *Stage 2: Consensus Formation* | | | | | | |
| *w/o* PF.-Agent | 0.702 | 47.85 | 0.535 | 54.05 | 71.65 | 69.85 |
| Uniform $\pi^m$ | 0.698 | 48.05 | 0.531 | 54.30 | 71.90 | 70.10 |
| *Objective terms* | | | | | | |
| *w/o* $L_{\text{gain}}$ | 0.698 | 48.10 | 0.531 | 54.35 | 71.90 | 70.10 |
| *w/o* $L_{\text{red}}$ | 0.703 | 47.70 | 0.536 | 53.95 | 71.55 | 69.75 |
| *w/o* $L_{\text{pub}}$ | 0.696 | 48.30 | 0.528 | 54.55 | 71.95 | 70.20 |
| only $L_{\text{task}}$ | 0.712 | 46.70 | 0.549 | 52.90 | 71.05 | 69.05 |

ble 3) isolate the contribution of each module by contrasting the full GCL framework against specific variants.

**Efficacy of Selective Interaction.** A critical hypothesis of this work is that indiscriminate information exchange can be detrimental due to noise propagation. The results strongly substantiate this premise. The Full exchange variant, which forces interaction across all valid pathways, exhibits the poorest performance among interaction strategies, yielding an MAE of 0.721 on MOSI. Notably, this is inferior to the only language baseline (0.714), indicating that unregulated fusion introduces more noise than information gain. In contrast, the complete GCL framework significantly outperforms both, validating that the performance gains stem from

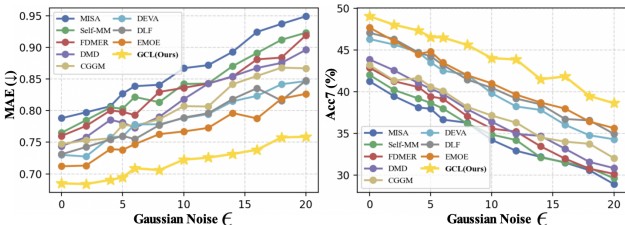

*Figure 2.* Robustness to Gaussian noise on CMU-MOSI. We inject additive Gaussian noise with varying standard deviation into all modalities. GCL demonstrates superior stability, maintaining the best MAE and Acc7 across all noise levels compared to baselines.

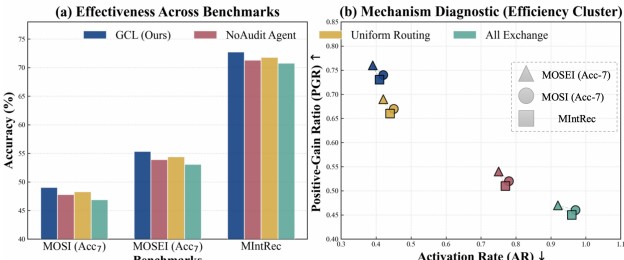

*Figure 3.* **Analysis of Audited Selectivity.** (a) task performance; (b) governance diagnostics plotting Activation Rate (AR) vs. Positive-Gain Ratio (PGR). GCL consistently occupies the high-efficiency quadrant by prioritizing beneficial exchanges, unlike baselines that suffer from excessive, low-gain communication.

the selectivity of the governance rather than mere connectivity. Furthermore, removing the Routing or Auditing Agent leads to consistent performance degradation. This confirms that the routing topology and the gain-based admission gate are mutually dependent; the former ensures topological sparsity, while the latter enforces semantic utility.

**Impact of Consensus Mechanisms and Regularization.** We analyze the contribution of the Stage 2 components and the auxiliary objectives. Removing the Public-Factor Agent sharply increases error (MAE rises to 0.702 on MOSI), suggesting that without an explicit shared factor, the model fails to disentangle common semantics from private channels, leading to feature redundancy. Similarly, relying on Uniform weighting in the Aggregation Agent degrades performance, confirming that sample-wise contribution awareness is essential for mitigating modality dominance. Regarding the training objectives, eliminating the redundancy control ($\mathcal{L}_{\text{red}}$) causes updated representations to collapse into co-adapted states, thereby diminishing the generalization capability. The most significant drop occurs when training with only $\mathcal{L}_{\text{task}}$, which emphasizes that the proposed structural governance is effective only when explicitly supervised by the combination of gain alignment, redundancy penalties, and public factor distillation. **Modality Complementarity.** The modality-specific ablations reveal the inherent hierarchy of information sources. While the Language modality serves as the primary semantic anchor, evidenced by the largest performance drop upon its removal, the superior performance of the tri-modal GCL over the best uni-modal variant confirms the successful integration of auxiliary acoustic and visual cues. Unlike naive fusion methods that may suffer from interference by weaker modalities, GCL effectively leverages these cues to resolve textual ambiguities.

### 4.4 Robustness Analysis under Gaussian Noise

To investigate GCL's resilience against signal corruption, we analyzed robustness on the CMU-MOSI dataset. We simulate sensor noise by injecting independent zero-mean additive Gaussian noise into the raw feature space of all modalities, varying the standard deviation from 0 to 20. As illustrated in Figure 2, while performance naturally degrades

across all methods as the signal-to-noise ratio deteriorates, GCL exhibits markedly superior stability compared to baselines. Existing methods relying on implicit fusion or static decomposition suffer precipitous drops in Acc7 and sharp increases in MAE. In contrast, GCL maintains a substantial performance margin, with its performance at severe noise levels often rivaling the baseline performance of competitors at moderate noise levels. The flatter degradation curve of GCL validates the efficacy of the proposed governed collaboration paradigm. When input features are corrupted, the Auditing Agent in the first stage detects a reduction in marginal predictive gain and actively gates off the transmission of noisy signals, thereby preventing the contamination of clean modalities. Furthermore, the Aggregation Agent in the second stage dynamically re-calibrates the contribution weights effectively down-weighting the private channels of severely corrupted modalities based on the consensus with the Public Factor. These results confirm that GCL does not merely memorize multimodal patterns but learns a robust decision boundary protected by explicit information control.

### 4.5 Generalization to Additional Modality Combinations

To examine whether GCL is tied to the original language-acoustic-visual sentiment setting, we further evaluate it on multimodal tasks with different modality compositions and task semantics, including acoustic-visual tasks and an image-text task. The governed interaction protocol is kept unchanged, while only the modality-specific encoders/features are adapted to each benchmark. As shown in Table 4, GCL consistently improves over competitive baselines on CREMA-D (Cao et al., 2014), UCF101 (Soomro et al., 2012), AVE (Tian et al., 2018), and Food101 (Bossard et al., 2014). These results indicate that the benefit of GCL comes from protocol-governed interaction and contribution-aware consensus, rather than from a dataset-specific design for sentiment analysis.

*Table 4.* Generalization to additional multimodal tasks with different modality combinations. We report Accuracy (%) and F1 (%).

| Methods | CREMA-D | | UCF101 | | AVE | | Food101 | |
| --- | --- | --- | --- | --- | --- | --- | --- | --- |
| | Acc. | Macro F1 | Acc. | Macro F1 | Acc. | Macro F1 | Acc. | F1 |
| MLA (Zhang et al., 2024) | 73.21 | 73.77 | 82.01 | 81.22 | 70.92 | 67.23 | 93.33 | 93.36 |
| D2R (Chen et al., 2024) | 73.52 | 73.96 | 82.11 | 80.87 | 69.62 | 64.93 | - | - |
| EAU (Gao et al., 2024) | - | - | - | - | - | - | 93.20 | 93.18 |
| ARL (Wei et al., 2025a) | 76.61 | 77.14 | 83.22 | 81.98 | 72.89 | 68.04 | 93.55 | 93.58 |
| **GCL (Ours)** | **77.46±0.07** | **77.97±0.06** | **83.95±0.15** | **82.75±0.08** | **73.64±0.10** | **69.06±0.12** | **93.75±0.18** | **93.78±0.15** |

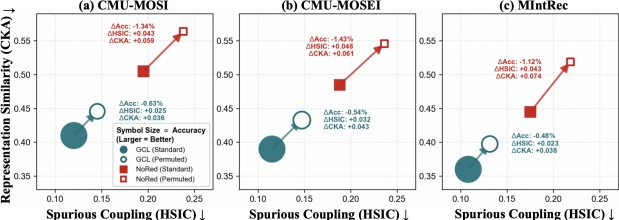

*Figure 4.* **Robustness against Spurious Coupling.** The axes denote coupling diagnostics (HSIC/CKA, lower is better) and symbol size represents task accuracy. GCL (teal) remains remarkably stable compared to the drastic collapse of NoRed (red), demonstrating that redundancy control effectively suppresses spurious coupling.

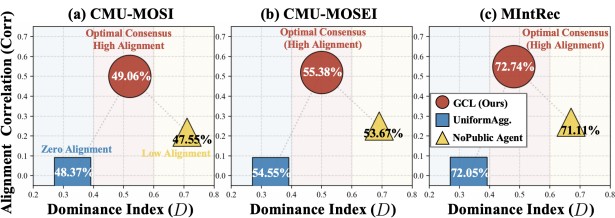

*Figure 5.* **Consensus Landscape Analysis.** Dominance Index ($D$) vs. Alignment Correlation (Corr); Symbol size represents task accuracy. While **UniformAgg** lacks adaptivity and **NoPublic Agent** degenerates into dominance collapse (high $D$, low Corr), **GCL** occupies the optimal equilibrium, maximizing alignment with genuine marginal utility while preventing modality collapse.

## 4.6 Further Analysis

**Analysis of Audited Selectivity.** This study investigates a fundamental question in multi-modal learning: does performance stem from the density of connections or the precision of governance? Hence, we contrast GCL with governance-relaxed variants (*NoAudit*, *Uniform Routing*, and *All Exchange*) and analyze the trade-off between the Activation Rate (AR) and the Positive-Gain Ratio (PGR). In Figure 3, while baseline variants exhibit a high AR, their corresponding PGR is markedly low, indicating that indiscriminate interaction introduces substantial noise and redundant coupling. Conversely, GCL achieves superior task performance across all benchmarks while maintaining a moderate AR and the highest PGR. This "efficiency cluster" confirms Stage 1's core ability to filter out non-beneficial exchanges. By strictly auditing the marginal predictive gain, the governance protocol ensures that cross-modal communication remains sparse yet semantically dense, converting ungoverned entanglement into high-precision, label-relevant cooperation.

**Robustness against Spurious Coupling.** To distinguish semantic causality from spurious shortcuts, we conduct a stress test using *Message Permutation*. By randomly shuffling sender messages within a mini-batch while preserving unimodal statistics, we deliberately inject mismatched co-activations to disrupt the established interaction topology. Figure 4 visualizes the system dynamics under this perturbation, utilizing HSIC and CKA as diagnostics for representational dependence and similarity, respectively. The results reveal a stark contrast: the trajectory of the full GCL framework (teal) exhibits remarkable stability, maintaining high

task accuracy with minimal coupling displacement. Conversely, the variant without redundancy control (NoRed, red trajectory) undergoes a catastrophic collapse, characterized by a sharp drift towards high dependence and significant performance degradation. This divergence empirically confirms that ungoverned interactions tend to amplify noise into brittle dependencies. In contrast, GCL's redundancy governance effectively enforces representational orthogonality, preventing the consolidation of spurious alignments and ensuring that the model generalizes beyond training set artifacts. See Table 6 (Appendix A) for details.

**Consensus Landscape Analysis.** To verify that the proposed consensus mechanism effectively mitigates modality dominance without compromising contribution alignment, we analyze the optimization landscape using two diagnostic metrics: the Dominance Index ($D$), which quantifies the concentration of aggregation weights, and the Alignment Correlation (Corr), which measures the consistency between the learned weights and the true marginal utility of each modality. As illustrated in Figure 5, the distinct clusters reveal the operational regimes of different strategies. The *UniformAgg* baseline resides in the suboptimal region with minimum dominance but zero alignment, reflecting a failure to adapt to evidence quality. More critically, removing the shared factor (*NoPublic Agent*) causes the model to drift significantly towards the high-dominance, low-alignment quadrant. This trajectory indicates a *dominance collapse*, where the aggregation degenerates into an over-reliance on the strongest raw signal rather than the most informative one. In contrast, GCL converges to the optimal equilibrium

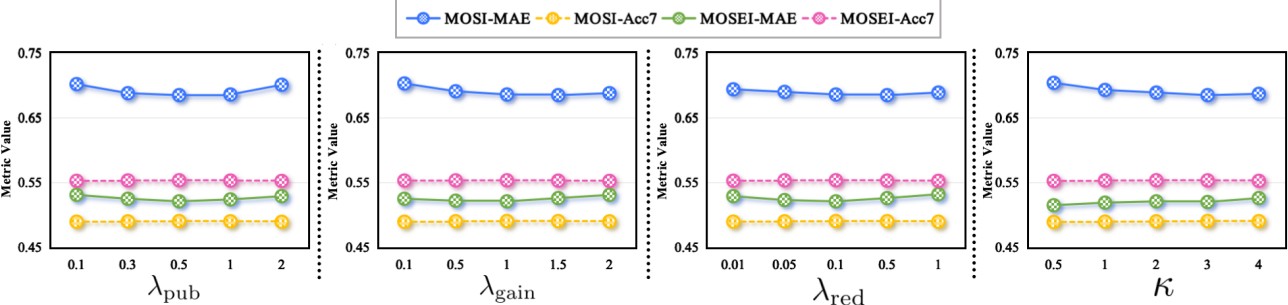

*Figure 6.* Sensitivity of GCL to $\lambda_{\text{pub}}$, $\lambda_{\text{gain}}$, $\lambda_{\text{red}}$, and $\kappa$ on the MOSI and MOSEI. Performance remains stable across values.

of high alignment and moderated dominance. This demonstrates that the Public Factor Agent successfully stabilizes the semantic decomposition, allowing the aggregation mechanism to assign weights based on genuine, label-relevant contributions rather than creating spurious shortcuts. See Table 7 (Appendix B) for complete experimental results.

**Efficiency Analysis.** We report GCL efficiency on CMU-MOSI in Table 5. GCL achieves superior Pareto efficiency. Specifically, compared to the ConFEDE (256.98M), GCL reduces parameters by 54% and latency by 50%. Against the recent MoE-based EMOE, GCL avoids structural redundancy, cutting parameters by 18% and training time by 25% (20.06s vs. 26.80s). This efficiency validates that our governed collaboration paradigm, relying on lightweight routing and auditing agents, is a far more efficient than the brute-force parameter expansion or complex expert stacking employed by current baselines.

*Table 5.* Computational efficiency comparison on CMU-MOSI. We report the total number of trainable parameters and the average training time per epoch.

| Model | Parameters | Time / Epoch |
|---|---|---|
| MISA (Hazarika et al., 2020) | 114.2 M | 24.18 s |
| FDMER (Yang et al., 2022) | 118.5 M | 29.5 s |
| ConFede (Yang et al., 2023) | 256.98 M | 40.12 s |
| EMOE (Fang et al., 2025) | 143.5 M | 26.8 s |
| **GCL (Ours)** | **117.56 M** | **20.06 s** |

### 4.7 Sensitivity Analysis

We evaluate hyperparameter sensitivity on CMU-MOSI and CMU-MOSEI by varying only the protocol critical controls that uniquely determine the governed interaction behavior of GCL, rather than auxiliary training knobs. Specifically, we study $\lambda_{\text{gain}}$ and the gate temperature $\kappa$ that regulate whether and how sharply edges are opened according to the teacher gain signal in Stage 1, together with $\lambda_{\text{red}}$ that calibrates redundancy suppression among specialized channels to prevent spurious co adaptation, and $\lambda_{\text{pub}}$ that stabilizes the public factor as the shared semantic anchor enabling contribution reweighting in Stage 2. These four hyperparam-

eters span the essential control axes of GCL, namely utility aligned sparsity, gating sharpness, coupling suppression, and anchor stability, and changing them yields interpretable protocol level failure modes, whereas varying supporting coefficients provides weaker evidence about the governance mechanism itself. As shown in Fig. 6, we vary one hyperparameter at a time and keep the others fixed, using $\lambda_{\text{pub}} \in \{0.1, 0.3, 0.5, 1.0, 2.0\}$, $\lambda_{\text{gain}} \in \{0.1, 0.5, 1.0, 1.5, 2.0\}$, $\lambda_{\text{red}} \in \{0.01, 0.05, 0.1, 0.5, 1\}$, and $\kappa \in \{0.5, 1, 2, 3, 4\}$, report MAE and Acc7. Across both datasets, GCL shows a broad stable region under moderate settings, with performance degrading only at extremes where interactions become overly permissive and admit dense low utility exchanges, or overly restrictive and suppress useful complementarity, while overly weak or overly strong redundancy and public supervision respectively increase coupling or reduce specialization and destabilize the shared anchor. Overall, these results indicate that the gains of GCL are not driven by fragile tuning but arise consistently from the proposed governance protocol.

## 5 Conclusion

We introduced Group Cognition Learning (GCL) , a governed collaboration framework to resolve modality dominance and spurious coupling. Unlike centralized fusion paradigms that risk overfitting to incidental correlations or gravitating towards the path of least resistance, GCL enforces a transparent, two-stage protocol to regulate information exchange. By implementing Selective Interaction, the framework utilizes routing and auditing agents to rigorously filter cross-modal communications based on marginal predictive gain, thereby suppressing redundant coupling. Furthermore, the Consensus Formation stage leverages a public-factor agent to disentangle shared semantics from private specialization channels, ensuring that the final aggregation reflects contribution-aware evidence rather than dominant signal biases. Experiments on regression and classification benchmarks show that this governance mitigates these limitations, achieves state-of-the-art performance, and supports more interpretable multimodal learning.

## Acknowledgements

This work is supported by National Key Research and Development Program of China (2023YFB4704100), National Natural Science Foundation of China (62201156), and Shanghai Municipal Science and Technology Major Project (No.2021SHZDZX0103). This study is also supported by (1) Shanghai Engineering Research Center of AI & Robotics, Fudan University, China, and (2) Engineering Research Center of AI & Robotics, Ministry of Education, China.

## Impact Statement

This work introduces a protocol-governed framework for multimodal learning, aiming to enhance the transparency and reliability of cross-modal interactions through explicit routing and auditing mechanisms. By transforming the opaque fusion process into a structured consensus formation, our approach mitigates the risks associated with spurious correlations and modality dominance, which are prevalent in conventional black-box fusion models. This methodological advancement offers potential benefits for building more robust and interpretable systems in domains requiring high-stakes decision support.

However, we acknowledge that the capability to extract fine-grained semantic consensus from multimodal data carries inherent ethical responsibilities. While the proposed governance mechanism is designed to filter noise and suppress redundancy, it could theoretically be repurposed to amplify specific, sensitive signal patterns if not properly constrained. We strictly advocate that this technology be utilized for improving system robustness in general-purpose applications rather than for surveillance or profiling individuals based on behavioral biometrics.

Furthermore, although our experiments on standard benchmarks demonstrate improved performance, the deployment of such models in open-world environments requires rigorous fairness audits. The selective interaction protocols, if trained on biased data, might inadvertently suppress minority modalities or underrepresented demographic features. Therefore, practitioners must ensure compliance with privacy regulations and conduct comprehensive bias evaluations before any real-world deployment. We encourage the research community to further investigate the long-term implications of governed interaction learning on algorithmic fairness and accountability.

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

## A    Detailed Numerical Results for Permutation Stress Test

Table 6 presents the complete quantitative results for the analysis in Section 4.6. While the main text visualizes the relative stability of GCL compared to the NoRed baseline, this table reports the exact absolute values for accuracy, HSIC, and CKA across all three datasets (MOSI, MOSEI, and MIntRec). It also quantifies the precise relative changes ($\Delta$), confirming that removing redundancy control significantly increases the model's sensitivity to spurious cross-modal correlations.

*Table 6.* Permutation stress test with coupling diagnostics and relative changes to GCL. $\Delta$ denotes (variant $-$ GCL). HSIC and CKA are averaged over modality pairs (lower is better). PM.$_*$ means PermuteMsg.

| Dataset | Metric | GCL (Ours) | NoRed | PM.$_*$+GCL | PM.$_*$+NoRed |
|---|---|---|---|---|---|
| **MOSI** | Acc-7 $\uparrow$ | 49.06 | 48.20 | 48.43 | 46.86 |
| | $\Delta$ | 0.00 | -0.86 | -0.63 | -2.20 |
| | HSIC/CKA $\downarrow$ | 0.120/0.410 | 0.195/0.505 | 0.145/0.446 | 0.238/0.564 |
| | $\Delta$HSIC/$\Delta$CKA | 0.000/0.000 | +0.075/+0.095 | +0.025/+0.036 | +0.118/+0.154 |
| **MOSEI** | Acc-7 $\uparrow$ | 55.36 | 54.40 | 54.82 | 52.97 |
| | $\Delta$ | 0.00 | -0.96 | -0.54 | -2.39 |
| | HSIC/CKA $\downarrow$ | 0.115/0.390 | 0.188/0.485 | 0.147/0.433 | 0.236/0.546 |
| | $\Delta$HSIC/$\Delta$CKA | 0.000/0.000 | +0.073/+0.095 | +0.032/+0.043 | +0.121/+0.156 |
| **MIntRec** | Acc/F1 $\uparrow$ | 72.74/70.95 | 72.02/70.10 | 72.26/70.30 | 70.90/69.10 |
| | $\Delta$Acc/$\Delta$F1 | 0.00/0.00 | -0.72/-0.85 | -0.48/-0.65 | -1.84/-1.85 |
| | HSIC/CKA $\downarrow$ | 0.108/0.360 | 0.175/0.445 | 0.131/0.398 | 0.218/0.519 |
| | $\Delta$HSIC/$\Delta$CKA | 0.000/0.000 | +0.067/+0.085 | +0.023/+0.038 | +0.110/+0.159 |

## B    Detailed Numerical Diagnostics for Consensus Formation

Table 7 provides the comprehensive numerical data for the analysis presented in Section 4.6. While Fig. 5 visually illustrates the trade-off between dominance and alignment, this table reports the exact values for Weight Entropy ($H(\pi)$), Dominance Index ($D$), and Alignment Correlation (Corr), alongside Task Accuracy for all three benchmarks. These results quantitatively confirm that GCL achieves superior performance by maintaining high contribution alignment, whereas removing the public factor leads to low-entropy dominance collapse.

*Table 7.* Detailed consensus diagnostics corresponding to Analysis III. $H(\pi)$ is the entropy of aggregation weights, $D = \max_m \pi^m$ is the dominance index, and $\mathrm{Corr}(\pi, \widehat{\varphi})$ measures alignment between weights and drop-one marginal utility.

| Method | MOSI | | MOSEI | | MIntRec | |
|---|---|---|---|---|---|---|
| | Acc-7 $\uparrow$ | $H(\pi)/D/$Corr | Acc-7 $\uparrow$ | $H(\pi)/D/$Corr | Acc $\uparrow$ | $H(\pi)/D/$Corr |
| GCL (Ours) | 49.06 | 0.99 / 0.52 / 0.50 | 55.36 | 1.00 / 0.50 / 0.52 | 72.74 | 1.02 / 0.48 / 0.55 |
| UniformAgg | 48.37 | 1.10 / 0.33 / 0.00 | 54.55 | 1.10 / 0.33 / 0.00 | 72.05 | 1.10 / 0.33 / 0.00 |
| NoPublic Agent | 47.55 | 0.78 / 0.71 / 0.22 | 53.67 | 0.80 / 0.69 / 0.25 | 71.11 | 0.82 / 0.67 / 0.27 |

