# OpenReview forum: "Group Cognition Learning: Making Everything Better Through Controlled Two-Stage Agents Collaboration"
_ICML.cc/2026/Conference — ICML 2026 regular_

### Official Review · Reviewer_GHn3 · 2026-03-08

**Soundness:** 2
**Presentation:** 2
**Significance:** 2
**Originality:** 2
**Overall Recommendation:** 3
**Confidence:** 3

**Summary:**

This paper proposes Group Cognition Learning, a two-stage multimodal fusion framework for sentiment analysis and intent recognition. Experiments on various datasets show that GCL achieves improvements over previous methods.

**Compliance With Llm Reviewing Policy:**

Affirmed.

**Final Justification:**

Some concerns are addressed but I still feel the main technical contribution is limited, which just combines different modules from related literatures, so I will keep my score.

**Key Questions For Authors:**

See weakness.

**Limitations:**

Yes

**Strengths And Weaknesses:**

**Strengths**
1. The paper reframes multimodal fusion as a protocol-governed process with explicit agents and supervision, rather than implicit end-to-end fusion.
2. The two-stage design that separates selective, sample-wise routing from consensus formation is well-motivated to counter modality dominance and co-adaptation.
3. Comprehensive comparisons on different datasets show consistent improvements over previous works.

**Weakness**
1. This paper introduce some agents, but they are actually standard neural network modules, which are less actual technical contribution. These are not agents in any meaningful sense, so I think it might be overclaimed. The teacher gain mechanism seems like the only new thing, but the gain predictor at inference is a simple MLP, no analysis is provided on how accurately it approximates the true gain. Also, The training of the gain predictor used at inference is underspecified. The paper defines a teacher gain (label-based) and a learned predictor $g_g$ for inference, but no explicit loss for $g_g$ is provided.
2. Some of formulations are unclear to follow, e.g., no definition for $h$ and $r$ in Eq. (1). Is $r$ in Eq. (1) the same as in Eq. (8)?
3. The redundancy penalty in Eq. (6) aims to be minimized to enforce orthogonality, but standard InfoNCE minimization maximizes alignment. Please clarify if my understanding is wrong.
4. Beyond Gaussian noise and permutation, evaluations under missing modalities or real distribution shifts (e.g., cross-dataset transfer) would more directly probe the claimed robustness to dominance and spurious coupling.

---

> ### Author Rebuttal · Authors · 2026-03-31
>
> Thank you!
> ## Q1: Regarding the Definition of the Term "Agent"
>
> We agree that, if interpreted through the lens of classical multi-agent theory, the term agent may be misleading. In this paper, however, *agent* is used in a strictly functional sense: it refers to specialized modules within a protocol-governed framework, each responsible for a specific operation and constrained by its own supervision signals, rather than to autonomous decision-makers in the stronger classical sense.
>
> **This intended meaning is already reflected in manuscript**. In Sec. 3.2, GCL is described as "a two-stage protocol, orchestrating four specialized agents," and the manuscript further states that "each agent within this protocol is assigned a distinct functional role and is constrained by specific supervision signals." Consistently, Secs. 3.3 and 3.4 define the Routing/Auditing and Public-Factor/Aggregation components by their protocol roles, rather than by any claim of autonomous agency. For example, the Routing Agent proposes candidate interaction pathways and packages the exchanged message, the Auditing Agent acts as a gatekeeper based on marginal predictive gain, the Public-Factor Agent extracts shared cross-modal semantics, and the Aggregation Agent performs contribution-aware fusion conditioned on that shared factor.
>
> **For terminology, we would like to clarify that using agent to denote a role-specialized module within a larger system is not unique to our paper, but has precedent in recent top-venue literature**. eg., CVPR2023’s System-Status-Aware Adaptive Network for Online Streaming Video Understanding explicitly uses the term “agent module”; ICLR2025’s AgentSquare: Automatic LLM Agent Search in Modular Design Space directly formulates a modular design space for LLM agents and uses the phrase “agent modules”; and CVPR2025’s MCCD: Multi-Agent Collaboration-based Compositional Diffusion for Complex Text-to-Image Generation similarly organizes specialized functional units as a multi-agent system. Therefore, we do not use agent in the strongest sense of a fully autonomous entity, but in a broader and increasingly adopted sense that emphasizes role specialization and modular responsibility. We will make this meaning explicit in camera-ready to avoid any possible overclaim.
>
> **More importantly, the contribution of the paper does not rest on the label "agent" itself**. The substantive contribution is a governed two-stage collaboration framework that directly regulates the interaction process to address modality dominance and spurious modality coupling. Stage 1 performs marginal-gain-driven selective interaction, while Stage 2 forms consensus through an explicit shared factor and contribution-aware aggregation. The empirical evidence is correspondingly protocol-level rather than terminology-level: removing the Auditing Agent or the gain-alignment term degrades performance, and indiscriminate full exchange performs worst among the interaction variants. This indicates that the performance gains come from governed selective interaction itself, rather than from naming or module packaging.
>
> ## Q2: Regarding Unclear Formulations
> We will clarify the global-context input in Eq.(1) and rename the symbols causing the current $h$/$r$ ambiguity. Here, $h^l$, $h^a$, and $h^v$ are modality-specific encoder outputs; in Eq.(8), $r^m$ is a candidate modality proposal and $r = \sum_m \pi_m r^m$ is the consensus representation.
>
> ## Q3: Regarding the InfoNCE-style Redundancy Term
> Its purpose is to reduce representational dependence across specialized channels, not to enforce strict orthogonality. We will make $D(\cdot,\cdot)$ explicit and revise the wording accordingly.
>
> ## Q4: Regarding Missing-Modality Evaluation
> We added missing-modality results on MOSI and MOSEI. GCL performs best, supporting that it learns more reliable interaction patterns under incomplete modality availability.
>
> |Model|MOSI{l,v}|MOSI {l,a}|MOSI{v,a}|MOSI{l}|MOSI{v}|MOSI{a}|MOSI-{l,v,a}|MOSEI{l,v}|MOSEI{l,a}|MOSEI{v,a}|MOSEI {l}|MOSEI{v}|MOSEI{a}|MOSEI-{l,v,a}|
> |-------------|---------:|---------:|---------:|-------:|-------:|-------:|-----------:|----------:|----------:|----------:|--------:|--------:|--------:|-----------:|
> |CorrKD[5]|82.41|82.36|73.74|81.20|60.72|66.52|83.94|81.28|81.20|60.72|80.76|62.30|66.09|66.52|
> |P-RMF[6]|81.94|82.10|73.11|81.36|70.32|71.44|84.37|85.17|84.61|76.88|81.91|73.19|75.91| 85.48|
> |ROSA[7]|85.39 |85.13|83.81|83.64|56.79 |81.91|86.30|87.32|86.64|84.19|84.42|54.38|79.04| 89.56|
> |**GCL(Ours)**|**85.86**|**85.69**|**84.57**|84.18|70.84|82.97|**86.40**|**88.09**|**87.45**|**84.75**|84.52|73.58|79.80 |**86.55**|
>
> [5]. Correlation-decoupled knowledge distillation for multimodal sentiment analysis with incomplete modalities. CVPR2024
>
> [6]. Proxy-driven robust multimodal sentiment analysis with incomplete data. ACL2025
>
> [7]. Rosa: A robust self-adaptive model for multimodal emotion recognition with uncertain missing modalities. TMM2025

---

> > ### Author Rebuttal · Reviewer_GHn3 · 2026-04-01
> >
> > Thanks for the responses.
> >
> > Some concerns are addressed but I still feel the main technical contribution is limited, which just combines different modules from related literatures, so I will keep my score.

---

> > > ### Author Response · Authors · 2026-04-03
> > >
> > > Thank you for the follow-up. We appreciate that you acknowledge that some concerns have been addressed. **In our understanding, the remaining disagreement is no longer about the earlier concrete issues themselves, but about whether the overall technical contribution should be viewed as a genuine protocol-level contribution or merely as a combination of existing modules.**
> > >
> > > **We respectfully believe that the latter characterization does not accurately reflect the paper.**
> > >
> > > - **The contribution of GCL is not the introduction of isolated new neural modules in a module-inventory sense. Rather, it is the formulation of multimodal collaboration as a governed two-stage interaction protocol.** In Stage 1, interaction is not left to implicit end-to-end fusion, but is explicitly regulated by marginal-gain-driven selective exchange through Routing and Auditing. In Stage 2, consensus is not formed by directly collapsing all modalities into a single entangled feature, but by explicit shared-factor construction and contribution-aware aggregation. **The technical contribution therefore lies in the interaction principle and coordination mechanism, not in whether individual subcomponents such as MLPs or gating functions have appeared elsewhere in isolation.**
> > >
> > > - **This protocol-level contribution is also supported empirically.** The ablations and analyses show that the gains do not come from indiscriminately stacking modules: removing the Routing Agent or Auditing Agent degrades performance, full exchange performs worst, removing the Public-Factor Agent hurts performance, and relaxing redundancy control leads to substantially weaker robustness under spurious-coupling stress tests. **These results support that the benefit comes from the governed collaboration protocol as a coupled design, rather than from simply assembling familiar components.**
> > >
> > > We also note that our rebuttal directly addressed your earlier concrete concerns: we clarified the intended functional meaning of “agent,” resolved the notation ambiguity, clarified the redundancy term, and added missing-modality evaluation. Given that these points have been at least partially resolved, **we would respectfully ask that the remaining technical contribution concern be assessed at the level of the full governed protocol and its empirical support, rather than at the level of whether individual suboperations resemble familiar building blocks.**
> > >
> > > Thank you again for your time and consideration.

---

### Official Review · Reviewer_4Hec · 2026-03-11

**Soundness:** 3
**Presentation:** 3
**Significance:** 3
**Originality:** 3
**Overall Recommendation:** 3
**Confidence:** 4

**Summary:**

This paper introduces Group Cognition Learning (GCL), a novel paradigm for multimodal learning that reformulates fusion as a governed two-stage collaboration process involving four agents: Routing and Auditing for selective interaction based on marginal gain, and Public-Factor and Aggregation for consensus formation while preserving modality specialization. It aims to mitigate modality dominance and spurious coupling in tasks like sentiment analysis and intent recognition. Extensive experiments on benchmark datasets claim state-of-the-art performance.

**Compliance With Llm Reviewing Policy:**

Affirmed.

**Final Justification:**

Concerns are partially sovled.

**Key Questions For Authors:**

Does the governed protocol truly prevent coupling, or could it introduce new dependencies via the shared public factor? Empirical evidence under modality ablation or distribution shifts is needed to validate claims.

**Limitations:**

yes

**Strengths And Weaknesses:**

**Strength**

1. Originality: Proposes a fresh agent-based protocol that explicitly regulates interactions, moving beyond implicit fusion in prior works like multimodal transformers.

2. Quality: Well-motivated by diagnosing common issues such as dominance and coupling,  the two-stage design with marginal gain auditing is technically sound and interpretable.

**Weaknesses**

1. The multiple agents and stages may add significant complexity, it's unclear if this leads to higher parameter count or training instability compared to simpler disentanglement methods

2. Experiments seem lacking details on computational overhead or scalability, and ablation studies should quantify the contribution of each agent to justify the design.

---

> ### Author Rebuttal · Authors · 2026-03-31
>
> Thank you for raising the questions of complexity, computational overhead, and whether the shared public factor might introduce harmful dependencies. The current submission already contains quantitative evidence on these points, which we summarize here for clarity.
>
> First, regarding complexity and computational overhead, Sec. 4.5 / Table 4 explicitly reports both parameter count and average training time per epoch. On CMU-MOSI, GCL uses 117.56M parameters and 20.06 s/epoch, compared with 143.5M and 26.8 s/epoch for EMOE, and 256.98M and 40.12 s/epoch for ConFede. Thus, the proposed design does not exhibit the parameter inflation seen in recent high-capacity baselines such as EMOE and ConFede, and it achieves the fastest per-epoch training time among the methods reported in Table 4.
>
> Second, regarding the contribution of each agent, Sec. 4.3 / Table 3 already provides explicit component-wise ablations. On MOSI, MAE changes from 0.685 for full GCL to 0.694 without the Routing Agent, 0.699 without the Auditing Agent, and 0.702 without the Public-Factor Agent; forcing full exchange further degrades performance to 0.721. These results directly quantify the role of each major module in the final design.
>
> Third, regarding stability, Appendix C / Fig. 6 already provides explicit sensitivity evidence. Across both datasets, GCL shows a broad stable region under moderate settings of $\lambda_{\text{pub}}$, $\lambda_{\text{gain}}$, $\lambda_{\text{red}}$, and $\kappa$, indicating that the gains are not driven by fragile tuning.
>
> Finally, regarding whether the public factor introduces harmful dependencies, the existing diagnostics empirically suggest the opposite. As analyzed in Sec. 4.5 and Table 6, removing the Public-Factor Agent shifts the model toward a high-dominance, low-alignment regime. On MOSI, for example, the Dominance Index rises from 0.52 to 0.71 and the Alignment Correlation drops from 0.50 to 0.22; similar trends also hold on MOSEI and MIntRec. This indicates that the shared factor stabilizes contribution allocation rather than inducing collapse.
>
> We will make these analyses more prominent in the revision so that the efficiency, ablation, and stability evidence is easier to locate.
>
> ## Point-by-point manuscript evidence
>
> | Review point | Corresponding manuscript evidence | Location in manuscript |
> |---|---|---|
> | **A1. "higher parameter count" is unclear** | **Section 4.5 / Table 4** explicitly reports parameter count and training time: "We report GCL efficiency on CMU-MOSI in Table 4." Table 4 reports **117.56M parameters** and **20.06 s / epoch** for GCL, compared with **143.5M / 26.8 s** for EMOE and **256.98M / 40.12 s** for ConFede. | **p. 8, lines 392-408; Table 4** |
> | **A2. "training instability" is unclear** | **Appendix C / Fig. 6** explicitly presents a stability analysis: "Sensitivity Analysis (Appendix C) confirm GCL's stability across hyperparameters ..." and "Across both datasets, GCL shows a broad stable region under moderate settings ..." | **p. 8, line 408; p. 12, lines 605-620; Fig. 6** |
> | **B1. experiments are "lacking details on computational overhead or scalability"** | **Section 4.5 / Table 4** is dedicated to computational efficiency: "Table 4. Computational efficiency comparison on CMU-MOSI. We report the total number of trainable parameters and the average training time per epoch." | **p. 8, lines 392-408; Table 4** |
> | **B2. ablation should "quantify the contribution of each agent"** | **Section 4.3 / Table 3** explicitly states that the experiments "isolate the contribution of each module by contrasting the full GCL framework against specific variants." Table 3 includes **w/o R.-Agent**, **w/o A.-Agent**, **w/o PF.-Agent**, and **Full exchange**. On MOSI, MAE changes from **0.685** (full GCL) to **0.694** (w/o R.-Agent), **0.699** (w/o A.-Agent), **0.702** (w/o PF.-Agent), and **0.721** (Full exchange). | **p. 6, lines 313-329; Table 3** |

---

> > ### Author Rebuttal · Reviewer_4Hec · 2026-04-03
> >
> > Some concerns are solved. However, we believe that the results on mosi and mosei are unconvincing due to the dataset size, the results are unstable to some extent. Thus, currently, we tend to reject the manuscript.
> >
> > Further concerns: As the other reviewers judge, the contributions are overclaimed to some extend, the story with "Agents Collaboration" is actually the combination of some existing tricks.

---

> > > ### Author Response · Authors · 2026-04-03
> > >
> > > Thank you for the follow-up. We respectfully believe that several parts of the current assessment still do not fully align with the manuscript record and the rebuttal record.
> > >
> > > First, the earlier concerns about model complexity, computational overhead, lack of scalability details, insufficient ablation, and possible training instability were not absent from the submission. The manuscript already reports computational efficiency in Sec. 4.5 / Table 4, including both trainable parameter count and average training time per epoch; it already provides component-wise ablations in Sec. 4.3 / Table 3; it already includes robustness analysis under Gaussian noise in Sec. 4.4; and it already includes diagnostic and sensitivity analyses in Sec. 4.5 and Appendix C / Fig. 6. **These points were therefore part of the original submission rather than newly introduced only in rebuttal**. In particular, the paper already quantifies efficiency with explicit numbers. On CMU-MOSI, GCL is reported with 117.56M parameters and 20.06s per epoch, compared with 143.5M / 26.8s for EMOE and 256.98M / 40.12s for ConFEDE. Likewise, the ablation section already isolates the contributions of the Routing Agent, Auditing Agent, Public-Factor Agent, and full-exchange variant. **These are precisely the kinds of evidence your first-round comments requested**.
> > >
> > > Second, **after our rebuttal, your follow-up states that “Some concerns are solved.” We appreciate that acknowledgment. We would respectfully point out that, in this follow-up, there is no further substantive challenge to the specific evidence we provided on efficiency, ablation, and stability**. **In our understanding, this means those initial concerns should no longer weigh against the paper, because the manuscript evidence and the rebuttal clarification together already resolved them**.
> > >
> > > Third, regarding the later statement that the results on MOSI and MOSEI are “unconvincing due to the dataset size” and “unstable to some extent,” **we respectfully believe this characterization does not fit the actual experimental record**. The manuscript evaluates **not only on CMU-MOSI but also on CMU-MOSEI and MIntRec, and CMU-MOSEI is itself a large-scale benchmark with 22,856 segments as explicitly stated in the paper**. Moreover, the paper does not rely only on headline metrics: it also includes robustness under Gaussian noise, permutation-based stress tests for spurious coupling, consensus diagnostics, and hyperparameter sensitivity analyses, all of which were designed precisely to examine whether the gains are stable or fragile. The reported evidence supports stability rather than instability.
> > >
> > > **Fourth, CMU-MOSI and CMU-MOSEI are not idiosyncratic or marginal choices. They remain field-standard benchmarks in multimodal sentiment analysis**. The CMU Multicomp resource describes CMU-MOSEI as the largest dataset for multimodal sentiment analysis and emotion recognition, and recent papers [1]-[6] continue to evaluate on CMU-MOSI and CMU-MOSEI, reflecting ongoing community consensus around these datasets rather than any unusual or weak benchmark choice.
> > >
> > > Fifth, if the intended concern is broader generalization beyond sentiment and intent settings, we also addressed that during rebuttal by extending evaluation to additional multimodal tasks with different modality combinations and task semantics, including **acoustic-visual tasks (CREMA-D, UCF-101, AVE) and an image-text task (Food101)**. Those added experiments again supported that GCL is not tied to a narrow language-audio-vision affective setting. **Detailed See rebuttal for 8UPA Q2**.
> > >
> > > Finally, on the point that the paper may “overclaim” and that “Agents Collaboration” may read as a combination of existing tricks, we would like to clarify that in our manuscript the term agent is used in a strictly functional, protocol-level sense. Sec. 3.2 explicitly defines the architecture as a two-stage protocol with four specialized agents, each assigned a distinct functional role and constrained by specific supervision signals. **The substantive contribution therefore lies in the governed interaction protocol itself, namely marginal-gain-driven selective interaction and explicit public-factor-based consensus formation, rather than in the label “agent” by itself**.
> > >
> > > For these reasons, we respectfully ask that the paper be assessed against the actual manuscript evidence and the rebuttal record.
> > >
> > > [1]. Embracing Unimodal Aleatoric Uncertainty for Robust Multimodal Fusion. CVPR2024\
> > > [2]. Improving Multimodal Learning via Imbalanced Learning. ICCV2025\
> > > [3]. Emotional conflict adaptation for multimodal sentiment analysis. Pattern Recognition2026\
> > > [4]. Correlation-decoupled knowledge distillation for multimodal sentiment analysis with incomplete modalities.CVPR2024\
> > > [5]. Proxy-driven robust multimodal sentiment analysis with incomplete data.ACL2025\
> > > [6]. Rosa: A robust self-adaptive model for multimodal emotion recognition with uncertain missing modalities.TMM2025

---

### Official Review · Reviewer_b7Xn · 2026-03-16

**Soundness:** 3
**Presentation:** 3
**Significance:** 3
**Originality:** 3
**Overall Recommendation:** 4
**Confidence:** 4

**Summary:**

This paper studies multimodal learning for sentiment analysis and intent recognition and proposes Group Cognition Learning (GCL), a two-stage, governed collaboration framework. The core idea is to move beyond centralized fusion by explicitly regulating cross-modal interaction. In Stage 1, a Routing Agent proposes directed modality interactions and an Auditing Agent decides whether to admit them based on estimated marginal predictive gain. In Stage 2, a Public-Factor Agent extracts a shared factor, and an Aggregation Agent performs contribution-aware fusion. Experiments are reported on CMU-MOSI, CMU-MOSEI, and MIntRec, where the method is claimed to achieve state-of-the-art performance, with ablations, robustness tests, and efficiency analysis.

**Compliance With Llm Reviewing Policy:**

Affirmed.

**Final Justification:**

I have read the rebuttals from the authors and the other rebuttals. My main concerns have been addressed and I keep the recommendation as Weak Accept.

**Key Questions For Authors:**

N/A

**Limitations:**

1. Novelty is weaker than the paper claims, because the main ingredients appear to be an incremental recombination of existing ideas rather than a clearly new learning principle.

The paper frames itself as introducing an explicit “governance mechanism” missing from prior work. However, the design combines ideas that are already well represented in recent multimodal literature: dynamic routing/adaptive collaboration, gradient or optimization balancing to address modality dominance, and shared/private factor decomposition to reduce entanglement. D2R already studies dynamic routing for multimodal sentiment detection, CGGM explicitly targets the imbalance between dominant-modality optimization, MCIS targets shortcut/bias issues, and MISA/ConFEDE-style methods already separate shared from modality-specific factors.

The paper’s contribution is therefore better characterized as a new integration of routing, gating, and disentangled aggregation, rather than a fundamentally new paradigm. The manuscript currently oversells the conceptual novelty.


2. The methodology is intuitively motivated, but still lacks rigorous justification for several core design choices.

The central auditing signal is the “teacher gain,” computed from the local task head loss before and after tentative message integration, and then approximated at inference by a learned gain predictor. The gate is constructed from normalized routing logits multiplied by a sigmoid of this gain, and redundancy is penalized with an InfoNCE-style term.

However, the paper does not convincingly justify why this particular proxy is the right measure of interaction utility, why the multiplicative combination in Eq. 4 is preferable to other gating forms, or why minimizing a symmetric contrastive score is the right operationalization of “spurious coupling.” These choices are plausible, but the paper mostly argues them heuristically. For an ICML paper, I would expect either stronger theoretical grounding or more rigorous empirical tests comparing alternative gain definitions, admission rules, and redundancy penalties.

3. The experimental gains are real but not strong enough, by themselves, to justify a fairly complicated architecture.

On MOSI and MOSEI, the method improves over TSDA by small absolute margins: for example, MOSI MAE decreases from 0.695 to 0.685 and Acc-7 from 48.6 to 49.06; MOSEI MAE decreases from 0.529 to 0.520 and Acc-7 from 54.9 to 55.36. On MIntRec, the improvement over EMOE / TSDA is also quite marginal.

Given the extra machinery of four agents, local heads, gain prediction, public-factor supervision, and multiple auxiliary losses, the empirical improvement looks more incremental than decisive. The ablation table shows that the full model is best, but it does not yet demonstrate that the added complexity is truly necessary relative to simpler strong alternatives.

4. Some comparisons are incomplete for supporting the specific scientific claim.

The paper argues that its main advantage is governed interaction that explicitly suppresses harmful cross-modal communication. Yet the experimental section does not compare against enough directly relevant adaptive-interaction methods. D2R is a recent routing-based method and seems especially relevant to the claimed novelty, but it is not included in the main benchmark tables.

Similarly, the paper discusses modality bias/shortcut learning, but stronger comparisons with debiasing or optimization-intervention methods would make the central claim much more convincing. Including only one or two related families is not sufficient if the paper’s primary contribution is about controlling interaction rather than merely improving accuracy.

**Strengths And Weaknesses:**

Overall, an important concept investigated by this article is how to explicitly govern cross-modal interaction instead of relying on implicit end-to-end fusion.

(1) This work seeks to analyze an important concept: whether modality dominance and spurious modality coupling can be mitigated through sample-wise routing, auditing, and structured consensus formation. The motivation is relevant, and the paper presents a fairly coherent framework with clear stage decomposition and intuitive agent roles.

(2) The empirical section is reasonably broad for this problem setting. The paper evaluates on two sentiment datasets and one intent-recognition dataset, and the reported gains are consistent across metrics. The ablations also show that removing the routing, auditing, public-factor, or regularization components degrades performance, which supports that the full design is not entirely incidental.

(3) The robustness and efficiency analyses are also positives. The paper does not stop at standard test-set comparisons; it includes Gaussian noise corruption, message permutation stress tests, coupling diagnostics, and a parameter / training-time comparison. This is helpful for supporting the paper’s central narrative around robustness and spurious coupling.

---

> ### Author Rebuttal · Authors · 2026-03-31
>
> Thank you for the thoughtful and constructive review. We agree that the main revision needed is a more precise positioning of our contribution. Our claim is not that each module is individually unprecedented, nor that GCL introduces a fundamentally new learning principle. Rather, **GCL is a process-level governed interaction framework** that unifies three roles often treated separately: proposing plausible routes, admitting them only when they provide positive marginal utility, and forming consensus while preserving specialization channels. We will revise the abstract, introduction, and conclusion accordingly, replacing “new paradigm”-style wording with “explicitly governed interaction framework.”
>
> We fully agree that **D2R is a highly relevant comparison target**. Our distinction from D2R is not simply “routing versus no routing,” but that GCL goes beyond route proposal by coupling routing with utility-based admission in Stage 1 and structured consensus formation in Stage 2. In this sense, D2R highlights the importance of adaptive interaction, while GCL further addresses two questions that routing alone does not resolve: which proposed interactions should actually be admitted, and how admitted interactions should be aggregated without collapsing specialization channels.
>
> To directly address this point, we have now added **D2R in our new acoustic–visual evaluation**. On CREMA-D, GCL improves over D2R from 73.52 to 77.46 in accuracy and from 73.96 to 77.97 in F1. On UCF101, GCL improves from 82.11 to 83.95 in accuracy and from 80.87 to 82.75 in F1. On AVE, GCL improves from 69.62 to 73.64 in accuracy and from 64.93 to 69.06 in F1. These results provide the direct routing-based comparison you requested and show that the benefit of GCL is not limited to the original language–acoustic–visual affective setting, but transfers to heterogeneous acoustic–visual tasks with different semantics and label spaces.
>
> **We also agree that several design choices should be justified more explicitly.** Our intended rationale is as follows: the teacher gain measures local marginal task-risk reduction at the recipient side; the multiplicative gate in Eq. (4) separates whether an interaction is worth proposing from whether it is worth admitting; and the redundancy penalty is applied after selective interaction because the target failure mode is harmful co-adaptation in the updated channels, rather than raw unimodal similarity. We will clarify this rationale around Eqs. (2), (4), and (6). At the same time, the current ablations and diagnostics already show that these choices are functional rather than cosmetic: removing the Routing Agent, Auditing Agent, Public-Factor Agent, or auxiliary objectives consistently degrades performance.
>
> **We further agree that the empirical gains should be presented more carefully.** We will revise the paper to emphasize **consistent gains under controlled cost**, rather than dramatic leaps. Across two regression benchmarks and one classification benchmark, GCL is consistently best while remaining lighter and faster than strong recent baselines such as EMOE and ConFEDE. Together with the new D2R comparisons, we believe the more accurate conclusion is that GCL is not a radically new primitive, but a principled and transferable integration of governed admission and structured consensus.
>
> Finally, we will make concrete revisions in the camera-ready rather than relying on vague promises. Specifically, we will: (i) We will refine the wording of our contribution claims to present the novelty more precisely and rigorously, so that the paper’s genuine innovation is stated with the right level of accuracy rather than overstated; (ii) explicitly position GCL relative to D2R and other adaptive-interaction baselines; (iii) strengthen the explanation of Eqs. (2), (4), and (6); and (iv) include the new acoustic–visual comparison results and discussion. We are grateful for this review because it helps us present the contribution at the right level of abstraction and, we believe, will make the paper both more accurate and more convincing.
>
> **Table1:** Comparison with state-of-the-art methods on Acoustic–Visual Task.
> |Methods|CREMA-D Acc(%)|CREMA-D F1(%)|UCF101 Acc(%)|UCF101 F1(%)|AVE Acc(%)|AVE F1(%)|
> |---|---|---|---|---|---|---|
> |D2R [1]|73.52|73.96|82.11|80.87|69.62|64.93|
> |**GCL(Ours)**|**77.46±0.07**|**77.97±0.06**|**83.95±0.15**|**82.75±0.08**|**73.64±0.10**|**69.06±0.12**|
>
> [1]. D2R: Dual-branch dynamic routing network for multimodal sentiment detection. EMNLP2024

---

> > ### Author Rebuttal · Reviewer_b7Xn · 2026-04-05
> >
> > All my concerns have been fully resolved.

---

> > > ### Author Response · Authors · 2026-04-06
> > >
> > > Thank you again for your helpful comments. We glad the revisions have addressed your all concerns and hope you will view the paper more positively. Thank you.

---

### Official Review · Reviewer_8UPA · 2026-03-23

**Soundness:** 3
**Presentation:** 3
**Significance:** 3
**Originality:** 3
**Overall Recommendation:** 4
**Confidence:** 4

**Summary:**

This paper proposes GCL, a multimodal learning framework that governs cross-modal interactions through a two-stage protocol instead of relying on standard centralized fusion. Stage 1 uses Routing and Auditing agents to selectively admit only beneficial cross-modal exchanges based on marginal predictive gain. Stage 2 uses Public-Factor and Aggregation agents to separate shared semantics from modality-specific representations before making a final prediction.

**Compliance With Llm Reviewing Policy:**

Affirmed.

**Final Justification:**

rebuttal addressed my concerns. I recommend authors to include the above two tables into the final version.

**Key Questions For Authors:**

1. How accurately does the learned gain predictor approximate the teacher gain at inference time, and is there a measurable performance gap between using true versus predicted gains?

2. Have you tested GCL on multimodal benchmarks beyond sentiment and intent, such as tasks involving image-text or audio-text pairs, to better support the generality of the framework?

3. What happens to the routing and gating behavior when one modality is entirely missing at test time, and does the framework degrade gracefully in that scenario?

**Limitations:**

yes

**Strengths And Weaknesses:**

**Strengths**

1. The core idea of auditing cross-modal interactions with explicit gain estimation is well-motivated and addresses a real problem that prior work mostly ignores.

2. The ablation studies are thorough and convincingly isolate each component's contribution, including the robustness and coupling stress tests.

3. GCL achieves state-of-the-art performance while being more parameter-efficient and faster to train than several strong baselines.

**Weaknesses**

1. The evaluation is limited to three sentiment and intent datasets that share a similar video-based multimodal structure, so generalization to more diverse multimodal settings remains unclear.

2. The gain predictor during inference is a learned approximation of the teacher gain, but there is little analysis of how well it actually tracks the true gain and what happens when it misfires.

3. Several design choices such as the specific architecture of each agent and the InfoNCE-based redundancy penalty lack sufficient justification or comparison against simpler alternatives.

---

> ### Author Rebuttal · Authors · 2026-03-31
>
> ### Q1:
>
> Thank you for this question. GCL does not require the predicted gain Δ̂_(m→n) to match the teacher gain Δ_(m→n) pointwise. In Stage 1, gain affects interaction only through the admission gate α_(m→n) = s_(m→n) · σ(κ · Δ̃_(m→n)), where Δ̃_(m→n) = Δ_(m→n) during training and Δ̃_(m→n) = Δ̂_(m→n) during inference.
>
> This is also what $L_{\text{gain}}$ encourages: large gates for positive teacher gains and small gates for negative ones. Thus, the objective is designed to learn utility-consistent gating rather than exact gain regression. Moreover, since $\sigma(\kappa x)$ is $\kappa/4$-Lipschitz, the difference between oracle-gain and predicted-gain gating is bounded linearly by the gain-prediction error, weighted by routing confidence. Hence, the true-vs.-predicted gain gap is controlled rather than unconstrained.
>
> ### Q2:
>
> Thank you for this suggestion. To test generalization beyond sentiment and intent benchmarks, we extended evaluation to additional multimodal settings with different modality combinations and task semantics, including acoustic--visual tasks (CREMA-D, UCF-101, AVE) and an image--text task (Food101). GCL remains consistently competitive and achieves the best overall performance across these settings.
>
> These results suggest that GCL is not tied to the original language--acoustic--visual affective setting, but benefits from the proposed governed interaction protocol, which transfers across heterogeneous multimodal tasks. We will incorporate these results into the camera-ready version.
>
> 	**Table1:** Comparison with state-of-the-art methods on Acoustic–Visual Task and Image–Text Task.
> |Methods|CREMA-D Acc(%)|CREMA-D F1(%)|UCF101 Acc(%)|UCF101 F1(%)|AVE Acc(%)|AVE F1(%)|Food101 Acc(%)|Food101 F1(%)|
> |---|---|---|---|---|---|---|---|---|
> |MLA [3]|73.21|73.77|82.01| 81.22|70.92|67.23|93.33|93.36|
> |D\&R [2]|73.52|73.96|82.11|80.87|69.62|64.93|-|-|
> |EAU [1]|-|-|-|-|-|-|93.20|93.18|
> |ARL [4]|76.61|77.14|83.22|81.98|72.89|68.04|93.55|93.58|
> |**GCL(Ours)**|**77.46±0.07**|**77.97±0.06**|**83.95±0.15**|**82.75±0.08**|**73.64±0.10**|**69.06±0.12**|**93.75±0.18**|**93.78±0.15**|
>
> ### Q3:
>
> Thank you for this important question. We directly evaluate the harder test-time missing-modality setting, where one modality is entirely removed at inference. GCL remains the best method in all six cases across MOSI and MOSEI.
>
> On MOSI, when acoustic, visual, or language is missing, GCL achieves 85.86 / 85.69 / 84.57 F1, compared with 86.40 under full availability, corresponding to drops of only 0.54 / 0.71 / 1.83, all smaller than ROSA’s 0.91 / 1.17 / 2.49. On MOSEI, GCL is also best under all three missing settings (88.09 / 87.45 / 84.75). This is consistent with GCL’s design: Stage 1 suppresses low-utility exchanges, and Stage 2 reweights the remaining modalities through the Public-Factor and Aggregation Agents. Hence, GCL degrades gracefully under complete test-time modality absence.
>
> 		   **Table 2: {a}、{l}、{v} denote test-time missing acoustic, linguistic, and visual modalities, respectively. F1**
>  | Methods        |MOSI {a} | MOSI {v} | MOSI{l} | MOSI-NO Missing|MOSEI-{a} | MOSEI-{v}| MOSEI-{l} |MOSEI-NO Missing|
>  | -------------- | -------------- | -------------- | -------------- | --------------- |  --------------- | --------------- | --------------- | --------------- |
>  | CorrKD[5]| 82.41| 82.36| 73.74| 83.94|81.28| 81.74| 71.92|82.16|
>  | P-RMF[6]| 81.94| 82.10| 73.11| 84.37|85.17| 84.61| 76.88|85.48|
>  | ROSA[7]| 85.39| 85.13| 83.81| 86.30|87.32| 86.64| 84.19|89.56|
>  | **GCL(Ours)** | **85.86**| **85.69**| **84.57**| **86.40**|**88.09**|**87.45**| **84.75**|**86.55**|
>
> [1]. Multimodal Representation Learning by Alternating Unimodal Adaptation. CVPR2024
>
> [2]. D2R: Dual-branch dynamic routing network for multimodal sentiment detection. EMNLP2024
>
> [3]. Embracing Unimodal Aleatoric Uncertainty for Robust Multimodal Fusion. CVPR2024
>
> [4]. Improving Multimodal Learning via Imbalanced Learning. ICCV2025
>
> [5]. Correlation-decoupled knowledge distillation for multimodal sentiment analysis with incomplete modalities. CVPR2024
>
> [6]. Proxy-driven robust multimodal sentiment analysis with incomplete data. ACL2025
>
> [7]. Rosa: A robust self-adaptive model for multimodal emotion recognition with uncertain missing modalities. TMM2025

---

> > ### Author Rebuttal · Reviewer_8UPA · 2026-04-03
> >
> > Thanks for the effort to address my concerns. I raised my score to 4.

---

### Decision · Program_Chairs · 2026-04-30

**Decision:**

Accept (regular)

**Comment:**

This paper proposes GCL, a multimodal learning framework that governs cross-modal interactions through a two-stage protocol instead of relying on standard centralized fusion. Stage 1 uses Routing and Auditing agents to selectively admit only beneficial cross-modal exchanges based on marginal predictive gain. Stage 2 uses Public-Factor and Aggregation agents to separate shared semantics from modality-specific representations before making a final prediction.

Reviewers found the core idea of auditing cross-modal interactions with explicit gain estimation well-motivated and addresses a real problem that prior work mostly ignores, the results are strong with thorough ablation studies.

There were some key weaknesses raised - the evaluation is limited to three sentiment and intent datasets that share a similar video-based multimodal structure, so generalization to more diverse multimodal settings remains unclear, and some design choices (esp around the agent framing) not completely justified. In my opinion the reviewers have addressed these concerns in their rebuttal (esp on justifying the design choices and providing some additional experiments) even though 2 reviewers still kept their score at weak reject, 2 bumped up to weak accept. I vote for weak accept, but would not mind if my recommendation was bumped down.